

# Investigating the surface mass balance of the Laurentide Ice Sheet during the last deglaciation

Kirstin Koepnick[1], Minmin Fu[2], and Eli Tziperman[1,3]

[1]School of Engineering and Applied Sciences, Harvard University
[2]Department of Earth & Planetary Sciences, Yale University
[3]Department of Earth & Planetary Sciences, Harvard University

**Correspondence:** Kirstin Koepnick (kirstinkoepnick@g.harvard.edu)

**Abstract.** In spite of decades of research, the role of climate feedbacks in the Pleistocene glacial cycles is still not fully understood. Here, we calculate the surface mass balance (SMB) of the Laurentide Ice Sheet (LIS) throughout the last deglaciation using the isotope-enabled transient climate model experiment (iTraCE). A surface energy balance framework is used to calculate yearly melt, and a parameterization of the refreezing of snow melt and liquid precipitation is incorporated. We compare the SMB to the ICE-6G reconstruction from the Last Glacial Maximum (LGM; 21 ka) until about 15–14 ka. We calculate the net ice mass rate of change from the difference between ICE-6G snapshots. The SMB calculated from iTraCE overestimates the net ICE- 6G ice mass loss rate, even ignoring ice flow and calving which would make the misfit even worse. We find the melt rate for the LIS to be primarily set by the small residual of large net shortwave and longwave radiative fluxes. The melt, and hence the SMB, are very sensitive to small changes in the albedo and downwelling longwave radiation. By increasing albedo by a mere 1.9% or by decreasing downwelling longwave radiation by only 1.45% (well within the uncertainty range of these variables), the large overestimation of the rate of mass loss deduced from the SMB compared to reconstructed rates of mass loss from 19–15 ka can be eliminated. The inconsistency of the climate model-derived, offline SMB calculation and the ice mass reconstructions exists irrespective of the role of ablation caused by ice flow, which cannot be calculated using this analysis. The extreme sensitivity of the melt rate suggests that General Circulation Models (GCMs) still struggle to reliably calculate the SMB, presenting a significant roadblock in our attempt to understand the Pleistocene ice ages.

## 1 Introduction

The retreat of the Laurentide Ice Sheet (LIS) following the last glacial maximum (LGM; 21 ka) left a lasting imprint on the North American landscape and is the most recent example of the response of ice sheets to warming. The history of this continental-wide ice retreat is reconstructed from uplift rates resulting from the continental isostatic adjustment (e.g., Peltier et al., 2015). This deglaciation occurred from approximately 21 ka to 12 ka (Clark et al., 2009) and involved significant climatic shifts such as Heinrich Stadial 1 (HS1), the Bølling-Allerød (BA) interstadial, and the Younger Dryas (YD) stadial. HS1 (18 ka to 14.5 ka) involved severe cooling and the discharge of massive icebergs into the North Atlantic (Hemming, 2004). The rapid warming brought by the subsequent BA (14.5 ka to 12.9 ka) period led to rapid glacial retreat. Meltwater pulse 1A (MWP-1A), occurring from 14.6 to 14.3 ka, led to a 10–20 m rise in sea level in less than 500 years (Fairbanks, 1989; Hanebuth et al., 2000;



Peltier and Fairbanks, 2006). However, this abrupt warm period was then interrupted by the YD stadial (approximately 12.9 ka to 11.7 ka) characterized by rapid cooling and a sudden return to glacial conditions (Broecker et al., 1989). The dramatic retreat of past ice sheets presents an intriguing analog for present-day warming. Specifically, the retreat of these ice sheets caused an average sea-level rise of approximately 1 meter per 100 years, a rate comparable to future climate change projections (Horton et al., 2014).

Variations in surface mass balance (SMB) and ice flow and discharge govern overall changes in ice sheet mass (van den Broeke et al., 2009; Khan et al., 2015; Fyke et al., 2018; Kapsch et al., 2021). The SMB is determined by the accumulation of snow, later compacted to form ice, and the loss induced by melting, evaporation, and sublimation. Meltwater and liquid precipitation either run off the surface of the glacier/ice sheet or percolate into the snow layer and refreeze (Pfeffer et al., 1991). In addition to the SMB, these large ice sheets are also affected by ice flow, calving, and basal melting, which together

account for the remainder of the mass loss during a deglaciation.

SMB is the residual of accumulation and net melt, whose spatial distribution and magnitude are largely affected by atmospheric and oceanic feedbacks and circulations. The presence of the LIS is thought to have profoundly influenced the atmospheric circulation and climate over North America (Kutzbach and Wright, 1985; Broccoli and Manabe, 1987; Löfverström et al., 2015; Liakka and Lofverstrom, 2018). As part of the coupled atmosphere-ocean system, these atmospheric changes certainly affected the SMB of the LIS (e.g., Roe and Lindzen, 2001; Löfverström et al., 2015; Liakka and Lofverstrom, 2018). The

effects of the LIS on ocean circulation have also been examined in a number of studies (e.g., Ullman et al., 2014; Zhu et al., 2014; Gong et al., 2015). Still, our understanding of the precise atmospheric and oceanic feedbacks that affect the evolution of continental-wide ice sheets during the ice ages remains incomplete.

Several studies have examined the SMB of ice sheets during and after the LGM using the outputs from atmosphere-ocean

general circulation models (AOGCMs). Carlson et al. (2009) explored the SMB of what was left of the LIS during the early Holocene ($\sim$ 9 ka) using an energy–mass balance model from Anslow et al. (2008) and found that the 9 ka LIS SMB contributes to a sea-level rise rate of $0.96 \pm 0.19$ cm/yr, consistent with geological records during that period. Using the same energy-mass balance model and refreeze scheme but with a different AOGCM, Carlson et al. (2012) investigated the response of the LIS to the Bølling warming and MWP-1A using a calculation of SMB from the AOGCM. Their study suggested that the ICE-5G

model (Peltier, 2004) might overestimate the LIS contribution to MWP-1A.

In another study, Ullman et al. (2015) found that the SMB of the LIS for key time slices, when forced by an AOGCM (Schmidt et al., 2014), was positive throughout much of the deglaciation, therefore suggesting that ice flow and dynamic discharge was mostly the cause of mass loss until about 9 ka. Moreover, this study implied that the sign of SMB is not a good predictor of glacial growth or decay. Conversely, Bradley et al. (2024) found that the SMB for the North American Ice Sheet

Complex (including the LIS) was negative during the LGM, pointing to a possible discrepancy between the AOGCM's SMB estimates and reconstructed ice volume history.

Kapsch et al. (2021) extended the analysis beyond specific time slices, examining the SMB and equilibrium line altitude (ELA), of various ice sheets during the last deglaciation. They used a transient simulation (MPI-ESM; Mauritsen et al., 2019) with different resolutions, combined with an energy balance model that included a parameterization for albedo evolution.



Kapsch et al. (2021) found that the SMB changes at the beginning of the deglaciation are a direct result of compensating effects of increasing accumulation and melt, specifically for the Greenland Ice Sheet (GrIS), with significant changes occurring around 14 ka, coinciding with the onset of the BA warming. Their study was the first to analyze SMB changes over the last deglaciation using a transient climate model forced with ice topographies from both GLAC-1D (Tarasov et al., 2012) and ICE-6G (Peltier et al., 2015). They focus on the SMB evolution of the GrIS, and show the evolution of the ELA of the LIS. Overall, there is still no consensus on whether the SMB of the LIS was positive or negative during the last deglacial period.

We calculate a time series of the SMB of the LIS utilizing climate model output from the isotope-enabled transient climate experiment simulation (He et al., 2021, hereafter iTraCE). This study, for the first time, calculates the SMB of the LIS during the *entire deglaciation* and compares the area-integrated SMB to ice mass loss rates using independent geophysical constraints as represented by ICE-6G (Peltier et al., 2015). Our analysis demonstrates the sensitivity of the SMB to small variations in albedo and downward longwave radiation and hence, the limitations of atmospheric models in calculating the SMB. We perform these offline SMB calculations from the iTraCE GCM output and incorporate a refreeze parameterization (Thompson and Pollard, 1997; Krinner et al., 2007; Colleoni et al., 2009) to account for the fate of surface meltwater or rain that percolates into the thermally active snow layer and is refrozen. While we focus on the SMB of the LIS, we find discrepancies between SMB and net ice mass loss rate from ICE-6G that cannot be explained by just including ice flow and calving because the SMB ice mass loss rate is larger than the net ice mass loss rate deduced from the ICE-6G snapshots. The time-continuous SMB and the comparison to ICE-6G allow for a more nuanced examination of SMB's role in ice sheet evolution and allow us to perform sensitivity tests of the variables that influence the SMB. By examining the potential sensitivities of SMB to climate parameters and testing the SMB's consistency with an ice volume history derived from geophysical isostatic constraints, we estimate the significance of SMB during different stages of the retreat of the LIS and evaluate the ability of state-of-the-art atmospheric models to provide accurate boundary conditions for dynamic ice sheet models.

This paper is organized as follows: we describe the methods used in section 2 where we first briefly summarize the ICE-6G reconstruction and describe the iTraCE experimental setup in section 2.1. We then describe the SMB calculation in section 2.2, report our results in section 3, and conclude in section 4.

## 2 Methods

### 2.1 ICE-6G Reconstruction & iTraCE Experimental Setup

The ICE-6G model (Peltier et al., 2015) reconstructs ice thickness history, including over the LIS, providing an estimate for the evolution of ice volume over the last deglaciation. This ice thickness and volume history are given on a $1° \times 1°$ global grid every 500 years. ICE-6G uses a glacial isostatic adjustment model in order to reconstruct ice volume history. This reconstruction does not incorporate ice flow dynamics or SMB and is not meant to be glaciologically self-consistent Tarasov et al. (2012). We compare the ice mass changes deduced from the SMB to the net mass changes calculated from changes between ICE-6G snapshots. ICE-6G was used for this purpose also because it is used as a boundary condition in the GCM simulations of iTraCE.





iTraCE is a transient GCM simulation of the last deglaciation performed using the isotope-enabled version of the Community Earth System Model version 1.3 (iCESM1.3 Brady et al., 2019). The iTraCE runs use the ice sheet boundary from ICE-6G (interpolated to the grid of the GCM), shown in appendix Figure A1. The horizontal resolution of the atmosphere and land components is 2° and the runs do not include a dynamically evolving ice sheet, although the prescribed ice-sheet boundary conditions are updated every 1000 years (except at 19 ka where 20 ka ice topography is used).

The iTraCE dataset includes four experiments that additively apply the four major changing boundary conditions. The first experiment (hereafter ICE) only involves changing ice sheets and ocean bathymetry (holding orbital conditions and greenhouse gas concentrations at 20 ka values). Next, insolation forcing from changing orbital conditions over the time period was added (hereafter ICE+ORB). Then, greenhouse gases (hereafter ICE+GHG+ORB) and meltwater fluxes were prescribed (referred to as ICE+GHG+ORB+WTR or simply iTRACE). We use "iTRACE" to refer to the fully forced simulation and "iTraCE" to reference the entire simulation suite. The iTRACE run is branched from the ICE+GHG+ORB run starting at 19 ka. This experimental setup is similar to Gregoire et al. (2015) who also investigated the contribution of greenhouse gases and evolving orbital parameters in the last deglaciation using a dynamical ice sheet model (GLIMMER-CISM; Rutt et al., 2009) forced by an AOGCM.

We take each monthly output variable, compute an annual average weighted by the number of days in each month, and then calculate the SMB using model outputs (an offline calculation) for each year using the method described below. iTRACE output is missing some years beyond 12 kyr, so we end our analysis at that point. For some of the analyses, we smooth the SMB time series using the Savitzky-Golay filter (Python package `scipy.signal.savgol`) with a window length of 100 years and a linear polynomial fit. For the purposes of this study, we only focus on the SMB and the evolution of the LIS.

## 2.2 Surface Mass Balance Calculation

To calculate the SMB of the LIS, we account for snow accumulation, sublimation, and refreeze of liquid precipitation and meltwater, as well as melting calculated via the energy balance at the surface. We do not consider the contribution to SMB from snow blowing/redistribution as it is not a model output of iTraCE. Over an ice sheet, CLM4.5 (Oleson et al., 2013) which is used in the iTraCE runs, assumes that sublimation occurs rather than evaporation, which we denote here as $E$. Snow accumulation is denoted by $P_S$ and liquid rain accumulation by $P_L$.

## 2.3 Melt calculation.

The surface melt, $M$, is calculated from the sum of radiative, turbulent, and geothermal heat fluxes (Vizcaíno et al., 2014). We calculate the melt rate $M$ (in m water equivalent (w.e.)/s) using the surface energy balance from Vizcaíno et al. (2014),

$$M = (SW_d(1-\alpha) - (\epsilon\sigma T_S^4 - LW_d) - SHF + GF)/(L\rho_w) - LHF/(L_s\rho_w). \tag{1}$$

The first term, $SW_d(1-\alpha)$, on the right-hand side of eq. (1) is the absorbed solar radiation where $SW_d$ is the downwelling shortwave radiation and $\alpha$ is the surface albedo. In the energy balance calculation, we set $SW_d(1-\alpha)$ to the net absorbed solar radiation variable from CLM. $LW_d$ denotes the surface downwelling longwave radiation; $\epsilon\sigma T_S^4$ is the upward surface



LW radiation, where $\epsilon$ is the emissivity of the snow/ice surface, $\sigma$ is the Stefan-Boltzman constant, $T_S$ the surface temperature;

The combination $\epsilon\sigma T_S^4 - LW_d$ is set to the net LW radiation at the surface from CLM. The term $SHF$ is the sensible heat flux; $LHF$ is the latent heat flux due to sublimation, and $GF$ is the geothermal heat flux. The use of geothermal heat flux in the SMB assumes that the ice sheet is in thermal equilibrium where the geothermal heat flux at the base is diffused to the surface rather than affecting the interior ice temperature. This quasi-thermal equilibrium assumption is needed because we do not model the thermal evolution of the ice sheet itself. Although geothermal heat flux is used, it hardly makes an impact on

overall changes in SMB. The results of not including $GF$ can be seen in Fig. A3.

The model variable names that are used in this analysis can be found in Table S1 in the appendix. Note that the sign convention of sensible and latent heat flux follows the CLM conventions. We set $GF = 0.05$ W/m$^2$ or $1.5 \times 10^{-6}$ m (w.e.)/s, the default value for the Community Ice Sheet Model (Lipscomb et al., 2019). Lastly, $L = 334$ kJ/kg is the latent heat of freezing/melting of water, $L_s = 2838$ kJ/kg is the latent heat of sublimation of ice, and $\rho_w = 1000$ kg/m$^3$ is the density of

water. These values are used to convert from W/m$^2$ to meters of water equivalent (w.e.)/s. Note that all variables used for the calculation of SMB are taken directly from Oleson et al. (2013). We break melt into its various components as shown in eq. (1) so that we can perform our sensitivity experiments shown in the following sections. Since SMB is not directly calculated in CLM4.5 (it is in newer versions of the model), we use all the shown variables to calculate melt rate and approximate refreeze so that we can calculate our offline SMB.

**2.4 Refreeze parameterization.**

Not all rain and meltwater runs off the surface of the ice sheet; a fraction $f$ is refrozen into the near-surface permeable layer of snow and firn. Based on Pfeffer et al. (1991), Thompson and Pollard (1997) proposed a parameterization that links $f$ to the ratio of annual snow or ice melt ($M$) to snowfall ($P_S$). This parameterization is then translated to a fractional refreeze given by Colleoni et al. (2009)

$$f = 1 - \min\left(1, \max\left(0, \frac{M/P_s - 0.7}{0.3}\right)\right), \tag{2}$$

where $P_S$ is the annually-averaged snow accumulation. This refreeze parameterization was originally used and developed for the Greenland Ice Sheet (GrIS). We assume it to hold for the LIS as well.

**2.5 SMB calculation.**

Finally, the offline SMB (using model outputs of iTraCE) is calculated in meters (w.e.) per second as in Vizcaíno et al. (2014)

$$SMB = P_s + fP_L - E - (1-f)M. \tag{3}$$

The SMB terms, therefore, include the snow accumulation, refrozen fraction of liquid precipitation, minus sublimation, and minus the part of the melt rate that is not refrozen. The SMB is multiplied by the number of seconds in a year to convert to units of m (w.e.)/yr.





Since some grid cells near the LIS margins contain fractional ice coverage (part ice, part land, or part ice, part ocean), we
take special care there. The radiative and turbulent heat fluxes for part ocean and part ice grid cells are properly weighted in
the CLM, implying that the flux represents only the ice fraction there. Given iTraCE output, we cannot separate ice fluxes from
land fluxes in LIS margin grid cells that are part ice part land. We, therefore, first calculate the area-integrated SMB excluding
these points, using only grid boxes that have an ice area fraction of one. We then calculate the SMB for grid cells containing
any fraction of ice and finally take the average of the two.

The motivation for this averaging is that the estimate using all grid cells that include full or fractional ice cover is an upper
bound on the magnitude of the SMB-derived LIS ice mass rate of change (red dashed line in Fig. 1a). Given the available iTraCe
model output, it is not possible to separate the fluxes that are responsible for the ice sheet warming/cooling rather than those
for the land in some LIS boundary grid points. For example, including these fractional grid cells might produce a relatively
high value of net shortwave radiation that, since the land has a lower albedo than the ice sheet, would artificially increase the
warming calculated from the surface energy balance in eq. (1). The land model already separates fluxes from ocean/land-ice
cells and so therefore, removing these fractional grid cells containing both ice sheet and land, provides a lower bound on the
magnitude of the SMB (dashed blue line in Fig. 1a) as it neglects some ice-warming fluxes in fractional ice-covered cells and
also slightly decreases the total surface area over which the SMB is integrated. The average of the two is an attempt to take
into account the warming and melting of the ice margins given the available AGCM output. We see from the dashed red lines
in Fig. 1a, omitting these margins decreases SMB by $\sim 1 \times 10^{12}$ kg/yr (w.e.).

The ICE-6G rate of change of the LIS ice mass, $M(t)$, shown in the figures below, is calculated as a centered finite difference
in time of the ICE-6G ice volume over the LIS $V(t)$ using

$$\frac{dM(t)}{dt}\bigg|_t = \frac{\rho_{ice}}{\rho_w}\frac{V(t+250\ yr) - V(t-250\ yr)}{500\ yr}, \tag{4}$$

where $V(t)$ in ICE-6G is given from 26 ka to 0 ka in 500-year increments. We multiply the finite difference by the ratio of the
density of ice, $\rho_{ice}$, and the density of water, $\rho_w$, to convert to an ice mass rate of change in w.e. units.

## 3 Results

In this section, we present time series of our calculated SMB (see section 2) compared to the ICE-6G rate of change of the LIS
ice mass from eq. 4. We then discuss how each component contributes to the SMB, which leads to an analysis of the SMB's
sensitivity to different variables, specifically albedo, downwelling longwave radiation, and snow accumulation. We then show
the spatial structure of the SMB and its various components.

The dark purple line in Fig. 1a is the smoothed yearly offline SMB calculated for the fully forced iTRACE experiment over
the LIS surface (accounting for ice margins as discussed in the methods). The shaded, light purple depicts the unsmoothed
yearly SMB, which fluctuates with a range of about $\pm 0.5 \times 10^{12}$ kg/yr around the smoothed SMB. As shown in Fig. 1a, the
SMB parameterization given by eq. (3) overestimates the ice mass rate of decline estimated by ICE-6G (black line, equation 4).
*The SMB indicates a rate of ice mass decline that is about four times faster than that of ICE-6G during 20 ka–16 ka.* This result





**Figure 1.** (a) Annual SMB for the LIS, integrated over the LIS surface and calculated using the full forced iTraCE experiment output (purple lines). The integrated SMB is compared to the ice volume rate of change of the LIS of ICE-6G (black curve). The solid, dark purple line depicts the smoothed annual SMB of the fully forced iTraCE experiment. The light purple line/shading depicts annual values, showing the variability of the SMB. The dashed red lines are the smoothed upper bound (calculated neglecting grid cells containing both ice and land grid cells) and the lower bound (calculated using all full and fractional ice-covered grid cells)–the purple line is the average of the two, see section 2. The thick black line depicts the ice volume change ($dV/dt$) calculated from the ICE-6G ice volume time series given every 500 yr (eq. 4). The light grey shaded region from 18 ka to 14.5 ka denotes the Heinrich Stadial 1 (HS1) time period. The light-yellow shaded region from 14.5 ka to 12.9 ka marks the Bølling-Allerød (BA) interstadial, and the light-shaded grey region from 12.9 ka to 11.7 ka marks the Younger Dryas (YD) stadial. We have limited the $y$-axis limit to show more clearly the SMB variability; however, note that $dV/dt$ of ICE-6G at 14 ka drops to approximately $-1.2 \times 10^{13}$ kg/yr (w.e.) and the full plot is shown by Fig. A2. (b) The colored lines depict the annual SMB of the other different experiments (blue: ICE, orange: ICE+ORB, green: ICE+GHG+ORB, purple: iTRACE). The solid-colored lines are the time-smoothed annual SMB, whereas the shaded regions are the annual values showing the variability in SMB. (c) The dark orange curve shows the adjusted SMB of iTRACE after increasing albedo by 1.9%. The green curve is the adjusted SMB after decreasing downwelling LW by 1.45%, and the blue, after increasing snow accumulation rate by 40%.



is different from that of Ullman et al. (2015), who calculated a positive SMB until about 9 ka using the ModelE2-R climate model (Schmidt et al., 2014) and ICE-5G ice sheet reconstruction (Peltier, 2004). Our results, however, are consistent with the findings of Bradley et al. (2024) who found negative SMB at the LGM.

Our SMB continues to exceed the ICE-6G rate of mass loss until around 14 ka, at which point the SMB first underestimates from 14 to 13.5 ka and then approximately tracks the evolution of ICE-6G. The period from 14 to 13.5 ka is when the LIS separates from the Cordilleran ice sheet (see Figure A1) and the onset of the Bølling-Allerød period (BA; 14.5 to 12.9 ka). Given the abrupt warming during the BA period, it is possible that ice flow and calving account for the difference between SMB and the estimated trajectory of the ice mass rate of change based on ICE-6G. If the SMB was less negative than ICE-6G (i.e., underestimated overall ice mass loss of ICE-6G), the residual between the two could be attributed to ice flow and calving.

On the other hand, a SMB more negative than ICE-6G ice mass change is a clear inconsistency. It may be indicative of either atmospheric model biases, inaccuracies in the calculation of the SMB, or an inconsistency with the ice height in ICE-6G and the climate system.

The other experiments in the iTraCE simulation suite are shown by the blue (ICE), orange (ICE+ORB), and green (ICE+GHG+ORB) lines in Fig. 1b. The ICE+ORB and ICE+GHG+ORB experiments show a similar long-term trend as the fully forced iTRACE

experiment. Changing orbital conditions make the biggest impact in both increasing the magnitude of SMB and affecting the overall trajectory. This is consistent with Gregoire et al. (2015) who found that orbital forcing caused 50% of North American ice volume loss during the deglaciation, greenhouse gases 30%, and nonlinear interactions between the two explained the remaining 20%. Note that they used positive degree days rather than a surface energy balance to calculate the melt rate.

The blue curve representing the ICE experiment shows a decreasing rate of ice loss over the integration period, deviating

from the general trend of the other three experiments. This experiment can be seen as strictly representing the internal ice-climate feedbacks which mostly involve ice albedo, atmospheric circulation, and the lapse rate elevation effect. Although evolving orbital parameters and greenhouse gases create the biggest difference and have an overall negative trajectory (Fig. 1), the internal ice-climate feedbacks also cause a large overestimation of SMB from 20 ka to about 15 ka supporting the idea that these feedbacks are important in deglaciation in addition to the effects of evolving orbital parameters and $CO_2$ (e.g.,

Milankovitch, 1941; Ganopolski and Calov, 2011; Abe-Ouchi et al., 2013; Heinemann et al., 2014). However, this conclusion may not be valid if a correction (e.g. increased albedo) to SMB were to be applied, thereby shifting the blue curves positive in Fig. 1b implying either orbital parameters are required for driving a deglaciation or that the SMB is not the major driver of the LIS deglaciation. During the BA, we see that the addition of rising greenhouse gases in the ICE+GHG+ORB experiment, given by the green curve, creates a larger magnitude of ice loss compared to the ICE+ORB experiment. The difference becomes

larger with the onset of the BA period and continues into the Younger Dryas.

The small difference between the ICE+ORB and ICE+GHG+ORB experiments is consistent with an amplifying and timing role for $CO_2$ in ice age evolution rather than a primary driving role (Abe-Ouchi et al., 2013; Gildor and Tziperman, 2000; Koepnick and Tziperman, 2024). Due to the increasing insolation over this period (Berger and Loutre, 1991), the abrupt warming from the re-invigoration of the Atlantic Meridional Overturning Circulation (Liu et al., 2009; Obase et al., 2021,

AMOC,) could have caused the increase in the ice mass rate of change in the ICE+ORB and ICE+GHG+ORB experiments





during the BA period. It is also possible that this warming occurred regardless of the cessation of glacial meltwater input (Obase and Abe-Ouchi, 2019). Regardless, the greater warming over the ice sheet during the BA appears to coincide with the increasing melt rate.

Overall, Fig. 1a illustrates three regimes in the evolution of SMB during the last deglaciation. In the first regime from 19 ka
to about 14 ka, the calculated SMB dramatically overestimates the total ice mass rate of loss compared to ICE-6G. Between 14 and about 13 ka, the SMB underestimates the large rate of ice loss, possibly indicating that ice flow dominates the total ice mass rate of change. Then, finally, between 13 and 12 ka, SMB provides a good estimate of total ice mass rate of change given by the ICE-6G reconstruction.

These results may imply that the SMB based on the atmospheric model's outputs is offset from the ICE-6G estimate by a
factor of 2–4 for larger ice sheets at the beginning of the last deglaciation, but is consistent with ICE-6G estimates for smaller ice sheets towards the end of the last deglaciation. The other experiments in Fig. 1b depict a similar trend except for ICE, which deviates from the trend showing only two regimes: one, dramatically overestimating SMB implying model or ice height errors and two, SMB underestimates overall ice evolution implying an ice flow dominant regime. The potential implied interpretation that the atmospheric model produces an accurate SMB for small ice sheets but not for large ones is speculative given the
sensitivity of the SMB that is explored below.

### 3.1   Averaged component analysis of the energy balance and the SMB.

To probe possible reasons for the fact that the SMB overestimates the ice mass rate of loss calculated from ICE-6G by a factor of 4, we plot the components of SMB for various time-slices (18.5 ka, 17.5 ka, . . ., and 12.5 ka) in Fig. 2a-c. Each bar shown is the century-mean of the labeled variable centered at the labeled age, averaged (rather than integrated) over the entire LIS (all
grid cells containing a full ice cover in the ICE-6G reconstruction). In Fig. 2a, the contribution of each term to the melt rate (eq. 1) for each time slice is shown. In this, the contributions of the shortwave and longwave radiation are about 8 times the magnitude of the net averaged melt rate (not accounting for refreeze), in agreement with the known fact that net shortwave and longwave are the main contributing factors to the overall melt rate (Oerlemans, 2010). Panel a also shows that the latent heat flux (due to sublimation) for each time slice is very small and contributes little to the overall melt rate. Comparing the time
slices as time progress from 18.5 ka to 12.5 ka, the magnitude of the averaged net shortwave and longwave decreases. Note that the components for this analysis are taken as an area average rather than an integral over the changing area of an ice sheet. This is done to make sure the ice area does not affect the conclusions regarding the physical balance behind the SMB. Thus the larger area of ice at 18.5 ka than 12.5 ka is *not* the reason for the larger contribution of the averaged net shortwave/longwave seen here. The large accumulation and melt relative to the net SMB suggests a sensitivity of the melt rate to relatively small
changes in each of the individual surface energy balance components affecting the melt rate of the ice sheet.

In Fig. 2b, we plot the century-average of each of the components contributing to the refreeze fraction calculated via eq. (2). The nonlinearity in the refreezing parameterization is evident here. As we move forward from 18.5 to 15.5 ka, there is a gradual decrease in melt rate and snow accumulation which is reflected in the decrease in the potential average area of refreeze. However, and 14.5 ka, both melt and snow increase yet the refreeze fraction still decreases. This increase in snow accumulation




is most likely a reflection of the increase in warming due to the onset of the BA period. The large increase melt rate at 13.5 ka and smaller increase in snow accumulation causes a large decrease in average $f$.

To understand how the components contribute to the overall SMB change between these time slices, Fig. 2c depicts the different terms in eq. (3). The largest magnitude change occurs at 13.5 ka reflected by the ice mass evolution of ICE-6G around the same time.

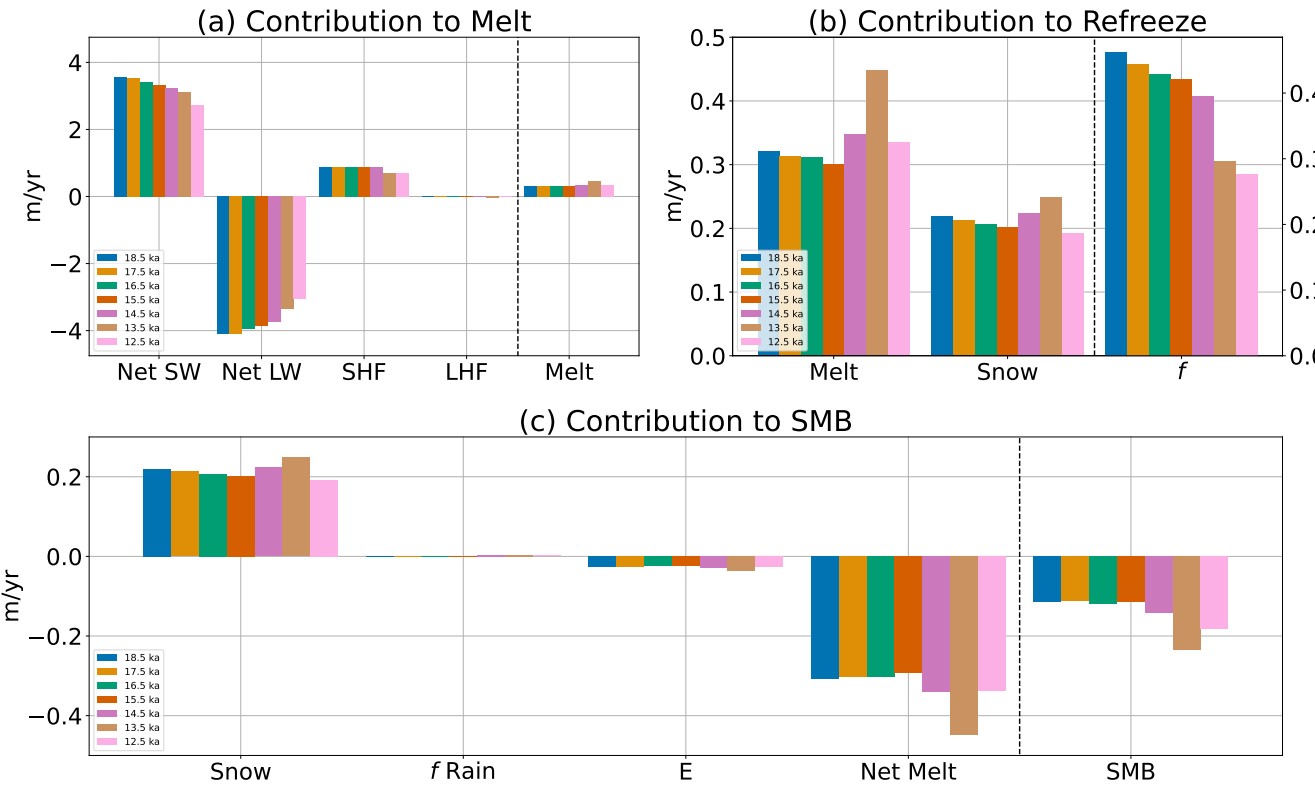

**Figure 2.** Bar plot of the components contributing to SMB during three different time slices (18.5 ka, 17.5 ka, . . . , 12.5 ka) during deglaciation. Each bar represents the century average centured at the labeled time slice of the separate components averaged over all grid cells containing full fractional ice coverage of the LIS. (a) The average contribution to melt from eq. (1). (b) The parameters involved in computing refreeze from eq. (2) and (c) the components contributing to SMB from eq. (3).

## 3.2 SMB sensitivity.


We find that small errors in the surface ice albedo and downward LW radiation calculated by the iTraCE/iCESM1.3 climate model can dramatically affect the SMB. To understand how sensitive SMB might be to a change in one of these input variables, we broke the net shortwave into its respective energy balance of $SW_{net} = SW_d(1 - \alpha)$ for downwelling shortwave radiation, $SW_d$, and albedo, $\alpha$. We solved for albedo using the iTraCE output variables for downward shortwave $SW_d$ and net absorbed



shortwave radiation, $SW_{net}$,

$$\alpha = (SW_d - SW_{net})/SW_d. \tag{5}$$

To see how much albedo might need to change to account for the overestimation of the ICE-6G ice mass loss rate between 19 ka and 14 ka, we introduced a factor $\beta$ to modify the albedo, writing the net absorbed SW as $SW_d(1 - \beta\alpha)$. We then set $\beta = 1.019$ thereby increasing albedo by only 1.9%. The value of $\beta$ was chosen to fit the fully forced iTraCE experiment to the ICE-6G ice volume rate of change from 19 ka to 18 ka.

The results of increasing albedo on annual SMB with $\beta = 1.019$ are shown in Fig. 1c by the thin orange line. The smoothed annual SMB, adjusted by the increased albedo, now fits the $dV/dt$ based on ICE-6G from 19 ka to 15.5 ka. While the albedo change was selected to allow the SMB to fit ICE-6G over 19–18 ka, the black and orange lines continue to match all the way to 15.5 ka, suggesting that perhaps this change is physically justified. The SMB then underestimates the rate of volume decline for the remainder of the time period, which is consistent with a larger role of ice flow and calving during that period.

Alternatively, we tried decreasing the net LW by introducing a factor, $\gamma$, as $\gamma(LW_d - \epsilon\sigma T_S^4)$ that is set to 0.9855 to fit the $dV/dt$ based on ICE-6G at 19 ka. Changes in cloud cover, type of clouds, etc. can also affect the amount of net absorbed longwave radiation. Accounting for these uncertainties is less straightforward. Cloud uncertainty can, in principle, be translated into an uncertainty of the atmospheric LW emissivity. Yet the downward LW comes from multiple atmospheric altitudes characterized by different emission temperatures and it is not possible to define a single emissivity and calculate the sensitivity to its value as we do in the following section for the albedo. Yet the uncertainty due to cloud emissivity still provides a possible alternative avenue to explain the discrepancy between the data and the discrepancy between the SMB and ICE-6G volume rate of change shown in Fig. 1a. Similarly to the sensitivity test with an increased albedo, the 1.45% decrease in downwelling LW also leads to a SMB that matches ICE-6G all the way to 15.5 ka, suggesting that this (e.g., cloud emissivity) might be another source of error in the difference between the ice mass rate of change in ICE-6G and SMB calculated from an atmospheric model.

Note that these sensitivity tests are conducted after model integration and do not include feedbacks. For example, increasing the albedo and thus decreasing the incoming SW will reduce the surface temperature. This, in turn, will decrease the upward LW radiation, acting as a negative feedback on the SW forcing. The result will be that the effect of the albedo change on the SMB will be damped. Given that we perform these calculations on model output, our goal is to identify the parameters that, with small changes, could have the most significant impact on SMB. Future work will need to re-examine this by including climate feedbacks: making such changes, and re-running the atmospheric model.

As one additional test, we increased snow accumulation by 40% such that, again, the SMB fits ICE-6G over 19 to 18 ka. This perturbed snow accumulation also tracks the ice mass evolution of ICE-6G until about 15.5 ka. This significant increase in snow accumulation needed to account for the difference in SMB and ICE-6G ice mass evolution suggests that snow accumulation may not be the model bias responsible for the difference between the ice evolution of ICE-6G and SMB. Another potential issue affecting the SMB/ICE-6G discrepancy is that the local height of the ICE-6G reconstructed ice sheet may be wrong, thereby affecting both the accumulation and ablation rates calculated by the atmospheric model. For example, Kapsch et al.





(2021) found that increasing resolution in their transient SMB model increased the accumulation in certain areas of the GrIS
due to the fact that the finer resolution model was able to better resolve complex topography. Such higher resolution, especially
around the steep margins, strongly influences orographically-induced precipitation as well as melt rates that are affected by
the surface ice temperature which, in turn, depends on the ice sheet height via the lapse rate. Since key dynamics like narrow
ablation zones occur on much smaller resolutions than the 2 or even 1-degree GCMs can resolve, insufficient resolution of
SMB can be addressed by statistical or dynamical downscaling (Lenaerts et al., 2019). Hanna et al. (2005) used atmospheric
reanalysis in combination with a meltwater retention model to retrieve SMB on a $5 \times 5$ km grid as an example of statistical
downscaling of SMB. Lenaerts et al. (2019) explain that statistical downscaling is challenging due to the complex and uncertain
dynamics of the atmospheric boundary layer on small scales, as well as due to the high small-scale spatial variability in the
surface albedo, roughness, and other characteristics. In a dynamical approach, a higher resolution regional climate model can
be used with reanalysis or GCM results as boundary conditions (e.g. Bromwich et al., 2004; Box et al., 2004; Fettweis et al.,
2013; Marshall et al., 2017), which would be out of the scope of this present study.

### 3.3 Spatial distributions of the SMB and its components.

We plot the spatial distribution of each of these components averaged over 100 years and centered at 14.5 ka in Fig. 3. Panels
(a–d) show the components that contribute to the melt rate shown in panel (e) as calculated via eq. (1). The net shortwave and
longwave radiation have the largest magnitudes. Sensible heat flux (SHF), shown in panel (c), is negative when it warms the ice
by sign convention. The refreeze fraction is shown in panel (g), and is a function of both the melt rate and the water-equivalent
snowfall rate shown in panel (f). The northeastern region of the LIS shows a refreeze fraction of 1, yet much of this area has
very little liquid precipitation or surface melting, if any, as can be seen in panels (i, k). Therefore, this refreeze parameterization
appears not to make much of a difference when computing net melt. Sublimation is shown in panel (j) and mostly occurs along
the south and south-eastern margins of the ice sheet. The net melt, after applying the refreeze fraction, is shown in panel (k)
indicating that most of the net melt occurs along the margins, as expected. Finally, the average SMB is shown in panel (l) which
is a combination of panels (f), (i), (j), and (k).

   The spatial structure of the SMB calculated for the iTraCE run at different times during the deglaciation is shown in Fig. 4
where each panel is the century-mean of each grid cell centered at the time noted in the title. As expected, we see more
ablation on the southwestern margin of the LIS and more accumulation on the western margin, due to orographically induced
precipitation incoming from the Pacific Ocean. The magnitude and spatial distribution shown in Fig. 4 are remarkably similar
to Fig. 4 of Kapsch et al. (2021), who used a simulation of the SMB, which uses a surface albedo evolution model, simulating
deglaciation using the Earth system model of the Max Planck Institute (Mauritsen et al., 2019). Although the spatial distribution
of SMB for the LIS is provided by Kapsch et al. (2021) for a few time slices, the full area-integrated SMB over time is only
provided for the GrIS, while our focus here is the LIS.

Basal melt rates are determined via a complex balance of geothermal heat flux from below, heat diffusion from the base into
the ice, frictionally-generated basal heat due to ice motion over the bed, and the advection of heat by the ice flow. Estimates of



**Figure 3.** Contour plots of the SMB components in the iTraCE experiment. Each subplot is the century average, centered at 14.5 ka, of the respective variable at each grid cell containing ice. The top row (a–d) shows the components that contribute to the melt rate in eq. (1), which is shown in panel (e). Panels (f–l) show other components that contribute to the overall SMB shown in panel (l), see text for details. Note that we do not show geothermal heat flux, which is set to a constant of 0.05 $W/m^2$.

the basal melt rates vary from 2 to 40 $mm\,yr^{-1}$ (XX references), significantly smaller than the SMB term considered above, allowing us to drop this contribution to a first order in the above analysis.



**Figure 4.** Contour plots of the SMB of the iTRACE experiment of the LIS with modern coastlines for comparison at different time slices during deglaciation. Each subplot is the century average, centered at the time slice noted in the respective title, of SMB of each grid cell containing ice. Red marks net ablation and blue marks regions of net accumulation. The contour plots of the SMB for ICE, ICE+ORB, and ICE+GHG+ORB are shown in the appendix.

## 4    Discussion

In this study, we estimated the surface mass balance (SMB) of the Laurentide Ice Sheet (LIS) using a surface energy balance to determine melt, and the fraction of surface melting and liquid accumulation that is refrozen into the snow layer. We take advantage of the existence of the transient simulation of the last deglaciation (He et al., 2021, iTraCE,), in conjunction with the estimate of the ice volume as a function of time given by Peltier et al. (2015). We use the two to test the consistency of SMB calculated from an atmospheric model with the ice mass rate of change calculated from geophysical isostatic constraints.

We find that the SMB significantly overestimates the total ice mass rate of loss from 20 ka until about 14.5 ka, by a factor of four, unlike Ullman et al. (2015) who found positive SMB over the last deglaciation using different climate model (Schmidt et al., 2014) and (ICE-5G, Peltier, 2004). This overestimation in SMB mass loss rate cannot be explained by including ice flow





or calving and, therefore, implies a discrepancy between climate models and paleo ice sheet reconstructions. iTraCE includes
several climate model runs, additively including the effects of ice thickness evolution based on ICE-6G (Peltier et al., 2015),
orbital parameters, greenhouse gases, and meltwater forcing. The SMB calculation overestimates the ICE-6G ice mass loss
rate for all experiments. We find that the addition of evolving orbital parameters makes the largest impact in changing the
magnitude and temporal behavior of the SMB over the last deglaciation (see Fig. 1b) in agreement with Gregoire et al. (2015).

To explain this discrepancy between ICE-6G and the iTraCE SMB and perhaps also the difference from the results of Ullman
et al. (2015), we analyzed the magnitude of each of the terms in the SMB at three different times (18.5 ka, 14.5 ka, and 12.5
ka) in Fig. 2a-c and the spatial distribution of the components at 14.5 ka in Fig. 3. The results (Fig. 2c) explicitly reveal that
the melt rate of an ice sheet is the small residual of large net shortwave and longwave radiation. As a result, slight errors in
either term lead to large SMB errors and contribute to the discrepancy between the ICE-6G and climate-model-derived, offline
SMB. We show, specifically, that small changes to the albedo (increasing it by merely 1.9%), which affects the net shortwave
radiation at the surface and, therefore, the melting rate, can reduce the discrepancy between the SMB and ICE-6G which cannot
be explained by ice flow and calving (see orange line in Fig. 1c). Similarly, we show that errors in downwelling LW radiation
at the surface (that could arise from errors in the simulation of cloud cover and their LW radiative effects) can dramatically
affect the surface energy balance. Increasing this downwelling surface LW radiation by only 1.45% (green line in Fig. 1c) can
make the SMB consistent with ICE-6G ice loss rate. On the other hand, snow accumulation needs to be increased by 40% for
the SMB to be consistent with ICE-6G (see the blue line in Fig. 1c), which implies that the SMB scheme used in this paper is
most sensitive to perturbations in the contributions to melt rather than to accumulation.

There are multiple caveats to note. As we have discussed in the context of Fig. 3c the iTraCE output does not allow the
separation of land fluxes from ice fluxes in grid cells along the ice sheet margins with fractional ice/land cover. We bound the
errors due to this issue and our results regarding the discrepancy of the SMB and ICE-6G are not affected by this. Addition-
ally, the margins of the ice sheet, where ice slopes can be higher, are difficult to resolve and accurately physically describe.
This prompted Kapsch et al. (2021) to compare the effect of two different resolutions (a coarser and a lower resolution) of
their transient simulation of the last deglaciation on SMB, finding that the lower resolution resulted in smaller differences in
accumulation rates between their results and a regional climate model. These results imply a tight link to topographic differ-
ences which might be smoothed by different resolutions and model biases in atmospheric circulation patterns which affect the
distribution of precipitation.
In addition, we use a refreeze parameterization (Colleoni et al., 2009) which was developed for the GrIS, whose applicability
to the LIS is not certain. This refreeze parameterization is also only a function of melt and snow accumulation and does not
include any explicit firn layer depth. If a refreeze parameterization that uses snow layer depth were to be incorporated into this
offline SMB calculation, special care of this thickness would need to be taken into account. Specifically, the land component
used in iTraCE (Oleson et al., 2013, or CLM4.5) places a cap on snow depth (w.e.) of 10,000 kg/m$^2$. Snowfall that tends to
increase this layer depth is assumed to be converted to ice and removed by ice flow and runoff. This so-called "snow cap"
represents ice flow and is not part of the SMB, which is our focus here. This cap is an important caveat to mention if a snow
accumulation term incorporated the depth of the annual snow layer instead of the accumulation rate.



| SMB Term | Variable/Value | Original Units | Model Component |
|---|---|---|---|
| $SW_d(1-\alpha)$ | FSA | W/m$^2$ | lnd |
| $LW_d - \epsilon\sigma T_S^4$ | FIRA | W/m$^2$ | lnd |
| $SHF$ | FSH | W/m$^2$ | lnd |
| $LHF$ | QSOIL $\times L_{subl}\rho_w$ | W/m$^2$ | lnd |
| $GF$ | 0.05 | W/m$^2$ | n/a |
| $P_s$ | SNOW | mm/s | lnd |
| $P_L$ | RAIN | mm/s | lnd |
| $E$ | QSOIL | mm/s | lnd |

**Table A1.** Data variables from iTraCE and the corresponding term from the SMB parameterization.

Another caveat to note is the effects of possible errors in ice sheet height in ICE-6G, which can affect both melting and accumulation rates calculated by the atmospheric model. It may be a useful avenue to compare the SMB produced by other ice history reconstructions (e.g., Tarasov et al., 2012; Lambeck et al., 2014) by rerunning iTraCE-like experiments for these reconstructions. In addition, the slope of ice sheet margins as well as the imposed roughness, can impact the gravity wave drag and affect the distribution of snow accumulation over the center of the ice sheet (Lofverstrom et al., 2020).

In summary, we find that a climate-model-derived, offline SMB predicts a much larger ice volume loss rate than the geophysical ice volume reconstruction of ICE-6G from 20 ka to about 15 ka, indicating a discrepancy that can only be explained by an error in one of these two estimates. The SMB is more consistent with the ice volume loss rate deduced from ICE-6G for later times, except during the BA period (14.5 ka to 12.9 ka), when the much faster ice loss rate of ICE-6G may suggest that ice flow rather than SMB dominates mass loss.

*Data availability.* The iTraCE output model data is available on the NCAR Casper /glade/campaign/cesm/collections/iTRACE and from the Climate Data Gateway at NCAR. The ICE-6G data is also available online at https://sites.physics.utoronto.ca/peltier_wr/datasets.

# Appendix A

To compute the weighted yearly mean, we take the month variable, $V_m$, and sum by the variable multiplied by the number of days in the corresponding month divided by total days in year $V_{ann} = \frac{\sum_m V_m T_m}{\sum_m T_m}$, where $T_m$ is the number of days in month $m$ summed from $m = 0$ to $m = 11$ (i.e. for an entire year).





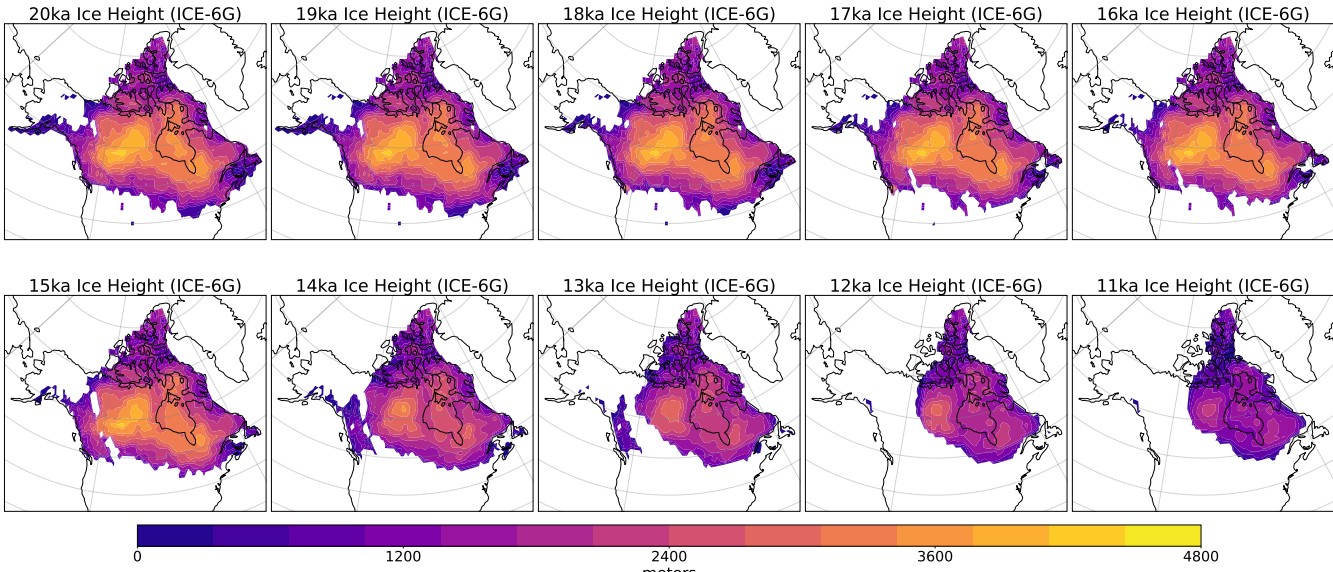

**Figure A1.** Ice height history of the LIS from 20ka to 11ka from ICE-6G given in meters with modern coastlines for reference.





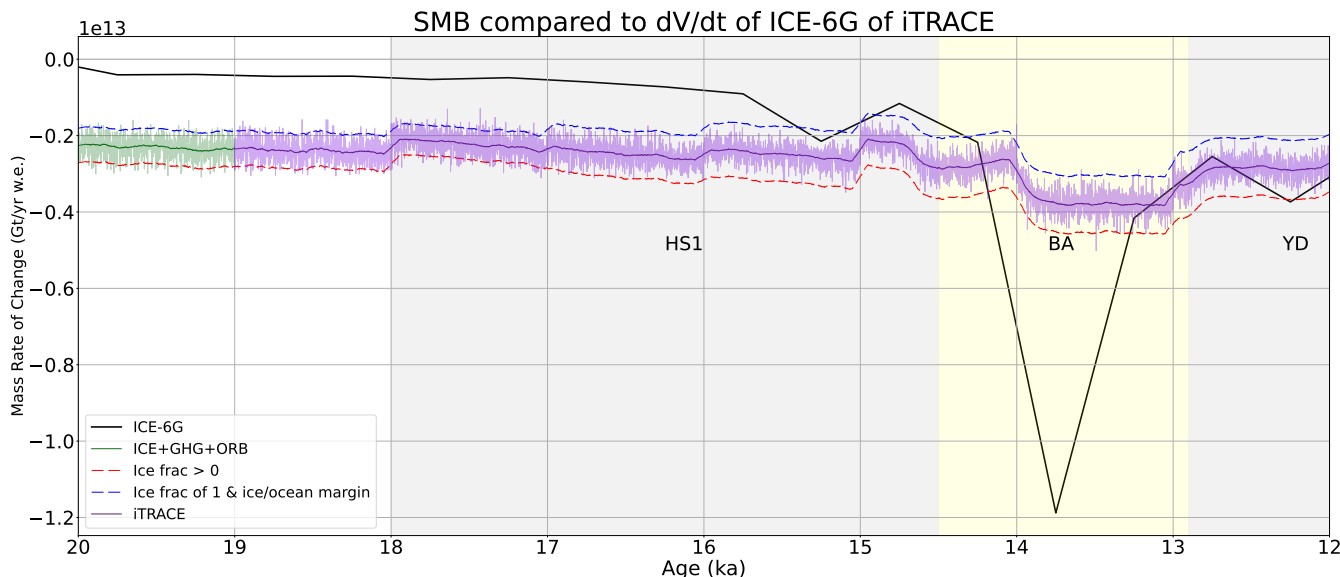

**Figure A2.** Annual SMB as in Fig.1 but without limit restrictions to see the full range of $dV/dt$ of ICE-6G.



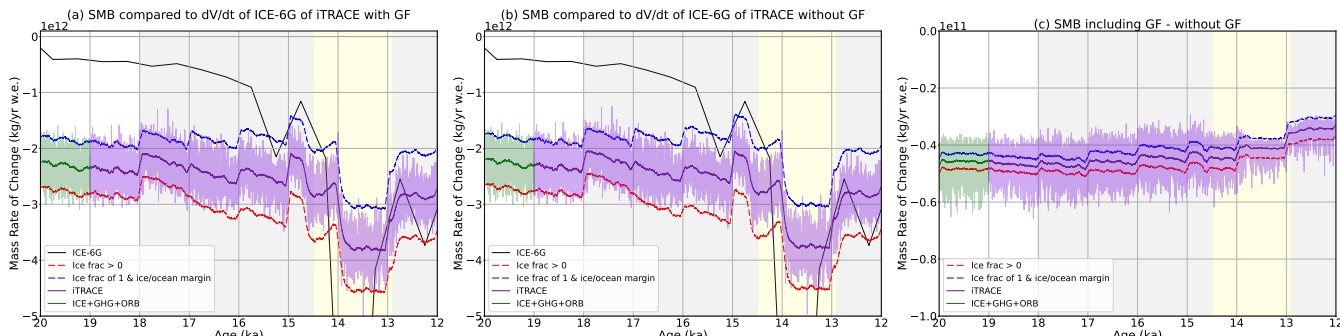

**Figure A3.** Annual SMB as seen in Fig.1 including geothermal heat flux (a), setting $GF = 0$ in (b), and the difference in annual SMB including $GF$ and not including.

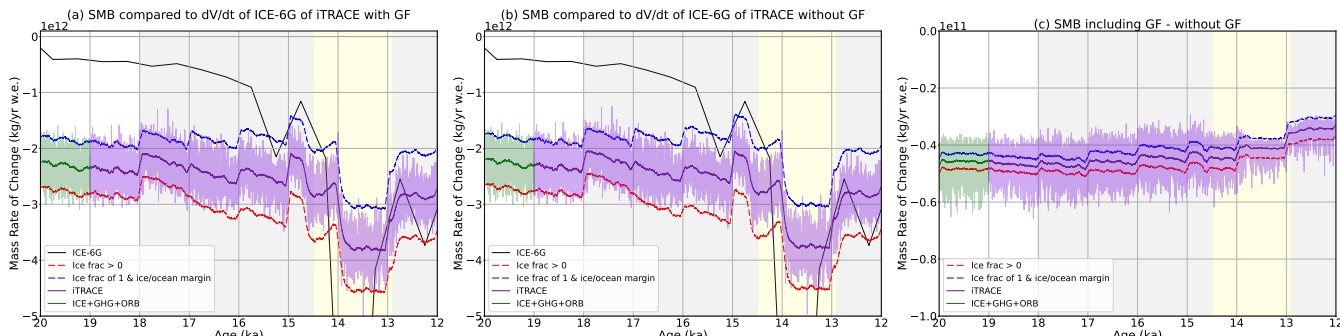

**Figure A4.** Spatial SMB similar to Fig.1, but for the ICE experiment.



**Figure A5.** Spatial SMB similar to Fig.1, but for the ICE+ORB experiment.





**Figure A6.** Spatial SMB similar to Fig. 4, but for the ICE+GHG+ORB experiment.

*Author contributions.* Kirstin Koepnick performed all programming/data analysis. Minmin Fu provided most useful discussions, advice, and
manuscript edits. Eli Tziperman overall advised the project in its entirety providing helpful next steps, debugging aid, and manuscript edits

*Competing interests.* The authors declare that no competing interests are present

*Acknowledgements.* We thank Bette Otto-Bliesner, Esther Brady, Robert Tomas, Zhengyu Liu, and Chengfei He, who produced iTraCE
and made it available to the community. We also thank Samuel Levis and Jiang Zhu for their help with interpreting various variables in
CLM/CAM. Lastly, we thank Jerry Mitrovica and Natasha Valencic for helpful discussions of ice volume reconstructions. This work has
been funded by NSF grant 2303486 from the P4CLIMATE program of the Geosciences (GEO) Directorate. ET thanks the Weizmann Institute
for its hospitality during parts of this work.



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
