# Peer review of "Investigating the surface mass balance of the Laurentide Ice Sheet during the last deglaciation"

_EGUsphere, 2024_

## Author Comment (AC1)

**Reviewer 1:**

General This paper presents estimates of the surface mass balance (SMB) of the Laurentide ice sheet (LIS) through its last deglaciation (21-12 ka ago). A benchmark is provided by the glacial isostatic adjustment (GIA) from ICE-6G, which puts thresholds on mass loss rates. An important discrepancy is found, which cannot be explained by the non-consideration of ice dynamics because the SMB model overestimates mass loss. Various potential reasons and sensitivities are discussed. The paper is interesting, well written and the figures are clear. The analysis is generally meaningful, but important information is lacking to get a clear picture of potential sources of error, see below.

Thank you for the summary of the work. We found your comments very helpful and have revised accordingly. Specially, we added a ground heat flux term into our energy balance, and this indeed made a significant difference on the resulting SMB. We also switched to a monthly melt analysis as suggested (the zero net surface energy balance when no melting occurs meant that this did not affect the SMB, but using annually-averaged values in the breaking of the SEB to different terms was indeed an error as you pointed out). We would like to express our gratitude for pointing out these two errors, and feel these comments and correcting these errors made our paper much stronger.

Major Comments:

l. 51: "In another study, Ullman et al. (2015) found that the SMB of the LIS for key time slices, when forced by an AOGCM (Schmidt et al., 2014), was positive throughout much of the deglaciation, therefore suggesting that ice flow and dynamic discharge was mostly the cause of mass loss until about 9 ka." This is inaccurate. When solid ice discharge is nonzero, SMB must be positive for an ice sheet in balance. When SMB decreases, but still remains positive, the ice sheet will lose mass. This is for instance the case for the contemporary Greenland ice sheet. Exactly as you state one sentence later "Moreover, this study implied that the sign of SMB is not a good predictor of glacial growth or decay."

We agree and revise the text to clarify, as follows:

> In another study, Ullman et al. (2015) found that the SMB of the LIS for key time slices (24, 21, 19, 16.5, 15.5, 14, 13, 11.5, and 9 ka), when forced by a GCM (Schmidt et al., 2014), was positive (that is, contributing the ice sheet growth rather than withdrawal) throughout much of the deglaciation. Ice flow could lead to a reducing ice mass even in the presence of a positive SMB, and Ullman et al. (2015) concluded that ice flow and dynamic discharge was indeed the main cause of mass loss until about 9 ka. This study therefore implied that the sign of the SMB is not a good predictor of ice sheet growth or decay.

l. 68: "...independent geophysical constraints as represented by ICE-6G". It is great

that these constraints are independent, but how accurate and suitable for this goal are they? One weakness is that GIA depends on the ice volume history used, and represents mass changes from both surface and ice dynamical processes.

Thank you for mentioning this. While ICE-6G may not be the best reconstruction and geophysical constraints may not be perfect, but AOGCMs have their own issues, and confronting the two should add value. In addition, ICE-6G is used by iTraCE and therefore, to be self-consistent, we chose to use it for this analysis. We have added the additional sentence to make this more clear:

> We acknowledge that geophysical ice reconstructions have their issues, including that they depend on ice volume history used and on uncertain assumptions regarding the isostatic processes involved. Yet, simulations of the SMB by climate models are, of course, far from perfect as well, and the comparison of the two therefore represents an interesting test of both approaches.

and

> Therefore, and for consistency with the iTraCE boundary conditions, we use ICE-6G (rather than other ice reconstructions, e.g., Tarasov et al., 2012) as an upper bound on the magnitude and a point of comparison for the SMB mass loss rate.

l. 120, Equation (1): (a) The equation contains the geothermal heat flux, but this flux is usually neglected in the surface energy balance as it is so small. Moreover, at the surface of a thick ice sheet the bedrock is even further away, further reducing this flux. You mention this later, but the text and even figure devoted to GF in l. 126-130 is too elaborate, given its insignificance (and uncertainty).

Thank you for this comment. We removed the geothermal flux, included instead the ground flux from iTrace. We now use the geothermal flux as a bottom boundary condition in our diffusion equation, which examines the sensitivity of the ground flux to refreezing latent heat release (Sections 2.4, 3.4).

l. 120, Equation (1): (b) What is however missing from the equation is the subsurface (or ground) heat flux, usually denoted by G. It is the conductive heat flux along temperature gradients just below the surface. This flux cannot be neglected and plays an important role in the modulation of surface melt, see e.g. a recent paper by Van den Broeke on the various energy contributions to melt in Greenland and Antarctica (doi: 10.1371/journal.pclm.0000203). As the only non-latent heat transport process, the subsurface heat flux also is important in the subsurface refreezing process. Why was this flux not included?

Thank you for mentioning this. We have now included the ground heat flux in our surface energy balance calculation, as well as provided a comparison of the CESM-computed to a diffusion equation model of the ground heat flux. We find that the

ground heat flux is indeed a critical part of the analysis, and including it led to significant changes to our results. We are grateful for this comment.

l.121-125: Radiative fluxes are the most important drivers of surface melt. By using the net surface radiative fluxes from CLM, you commit to CAM's radiation schemes and CLM's (snow/ice) albedo scheme. Please describe these schemes here. What albedo values are used for ice and snow, are impurities considered, impact of clouds and snow wetness/grain size etc.? The modern CLM has an elaborate snow albedo scheme.

We have added a discussion of the radiation scheme as follows,

> The atmospheric model used in iTraCE, the Community Atmospheric Model (CAM5, Neale et al., 2012) uses the Rapid Radiative Transfer Method parameterization (RRTMG, Iacono et al., 2008; Mlawer et al., 1997), which uses a correlated k distribution method for calculating radiative fluxes and heating rates. RRTMG is a widely used radiative transfer code, and shows improvements in its agreement with line-by-line radiative calculations compared to the older CAM radiation package (CAMRT, Neale et al., 2012).

And the snow albedo values:

> The Snow, Ice, and Aerosol Radiative Model (SNICAR) is used to simulate snow albedo and the absorption of solar energy within individual snow layers. The snow albedo is influenced by the solar zenith angle, IR band, the albedo of the surface beneath the snow, concentrations of aerosols deposited from the atmosphere (e.g., black carbon, mineral dust, and organic carbon), and the effective grain size of ice ($r_e$), which is modeled through a snow aging process (Oleson et al., 2013; Flanner and Zender, 2005; Flanner et al., 2007). Glacier albedos are set to 0.80 and 0.55 in the visible and near-infrared, respectively (Lipscomb and Sacks, 2012).

Section 2.3: Other important information is missing in this section. What was the time step used for the melt calculation? Many processes associated with melt over polar ice caps are highly nonlinear, so using e.g. daily averages of energy fluxes to calculate melt will lead to large uncertainties, especially in regions where melt is non-continuous.

The iTraCE dataset has monthly outputs, and following both reviewer's suggestions, we now calculate melt on a monthly basis. We then take annual averages of snow accumulation and melt to be used for the refreeze parameterization. when presenting the surface energy budget and surface mass budget (SEB,SMB) analysis, we average over months in which melting occurs only to show the resulting annual averaged fluxes. Please see our revised Eqs. (1), (5).

l. 145, Equation 2: Upon refreezing, large amounts of latent heat are released in the snow/firn. This will reduce the subsequent refreezing capacity. Is this accounted for?

Thank you for this important comment. To our knowledge, refreezing is not included in iTraCE and the refreezing latent heat release is therefore not accounted for in the CESM computed ground heat flux. However, following this comment, we added a diffusion equation analysis and perform a sensitivity test by including a parameterized latent heat of refreezing term into our diffusion model of the ground heat flux in our results section. We find that the latent heat release due to refreeze does not substantially affect the results of the diffusion model. Please see Sections 2.4, 3.4 and Figure 5.

l. 150, Equation 3: this way of defining SMB includes refreezing or 'internal accumulation' and is formally referred to as 'climatic mass balance' (see glossary of mass balance here: https://wgms.ch/downloads/Cogley_etal_2011.pdf). Fine to define SMB this way (many do it) but for clarity it's good to show that you're deviating from the formal SMB definition.

Thank you for mentioning this potential point of confusion. We have added the following:

> Note that by including refreezing or "internal accumulation", we are formally modeling the "climatic mass balance" (Cogley et al., 2011) and are therefore choosing to deviate from the formal definition of SMB. Although, as shown later, the inclusion of this specific refreeze parameterization has a very small effect on overall ice sheet mass balance.

Same line: I find the notation of the mass fluxes confusing. If 'P' stands for precipitation, I interpret $P_s$ as solid precipitation (snowfall) and $P_l$ as liquid precipitation (rainfall). The SMB equation then becomes, with $P_s - SU$ being snow accumulation: $SMB = P_s + P_l - SU - RU$ where SU is sublimation and RU is runoff, which can be written as $RU = (ME + P_l)(1 - f)$ where $f$ is the refrozen fraction. Substitution gives $SMB = P_s - SU + fP_l - (1 - f)ME$. Cautionary note: you use 'SMB' both for ice sheet integrated mass change (kg/yr, Fig. 1) as for specific mass loss (m/yr, Fig. 2). Avoid the term 'net melt', instead use runoff or the likes.

Thanks for pointing this out. We changed the presentation to first show the surface energy budget and then use that for the surface mass balance. We hope the revised presentation is clearer. whenever using "net melt" we add (runoff) to clarify, following this suggestion.

Figure 2: Are these fluxes averaged during melt?

Yes, the fluxes contributing to melt are now averaged during melt months only.

Figure 3: Not sure if I understand the signs of all these fluxes. Should netLW not be negative and SHF predominantly positive? In meteorology, SHF is defined positive

when warming the surface.

We revise all fluxes to be positive toward the surface. That is, atmospheric fluxes are positive downward, and the ground flux is positive upwards, again toward the surface.

> All fluxes are defined as positive downwards toward the ice surface, except the ground heat flux $GHF$ which is defined as positive upwards (that is, again positive towards the ice surface).

Minor and textual comments:

l. 3: "...the isotope-enabled transient climate model experiment (iTraCE)." Is the fact that the model is isotope enabled relevant for this work? If so, please state that here and explain why. Also applies to l. 66.

The isotope part is not relevant for this work, and we now mention this explicitly.

> While this version of the model also simulates water isotopes, this study uses only the physical climate variables from the simulation.

Figure A1: Please include ice thickness over Greenland also.

We now show the ice thickness over Greenland in Figure fig:ice-6g, as suggested.

l. 115: "Snow accumulation is denoted by PS". Do you mean snowfall? Snow accumulation is usually defined as snowfall minus sublimation.

Thank you, we made sure to use accumulation rate or snowfall in the appropriate contexts.

l. 116: "liquid rain accumulation". This is unclear, rain is always liquid and rain does not tend to accumulate. Did you perhaps mean rainfall?

Yes, thank you for mentioning the lack of clarity with this. We changed to be "rainfall".

Figure 1: In y-axis labels adding "w.e." is not relevant, because you provide integrated mass fluxes in kg/yr. On the other hand, in Fig. 2 (units m/yr) adding 'w.e.' is relevant, but not done...

Thank you for pointing this out. Corrected.

Figure 1: How deep does the ICE-6G curve dip below the x-axis in the BA? Ah, I see this is presented in Fig. A2.

Yes, it drops to approximately $-12 \times 10^{15}$ kg/yr and this value is now noted in the caption.

Reference list: the reference list was messy and hard to read, as it contained a mix of names with/without first names and did not start with last names.

Thank you for pointing this out. We have now fixed it.

l. 192: "Given the abrupt warming during the BA period, it is possible that ice flow and calving account for the difference between SMB and the estimated trajectory of the ice mass rate of change based on ICE-6G." Would be good to mention some processes that explain why strong warming could lead to enhanced ice flow and calving.

Thank you for mentioning this. We have added the following after the text that was at Ln. 192 in the previous version:

> It is possible that this abrupt warming during the BA period triggered enhanced ice flow and calving and a regime change from mass loss due to SMB to ice flow-dominant mass loss. Overall, the negative sign of the SMB is consistent with the decay of the LIS (note that this is not the case for the present-day Greenland Ice Sheet). Enhanced surface melting can lead to stronger injection of water into the base of the ice sheet via moulins (Colgan and Steffen, 2009; Banwell et al., 2016). This can accelerate the ice flow and calving, consistent with the fact that our calculation indicates stronger effect of ice flow and calving later in the deglaciation, where the SMB indicates more melting. While increased surface melting associated with negative SMB during a deglaciation can accelerate ice flow and enhance calving, the ice dynamics are also influenced by internal properties and subglacial interactions (Golledge et al., 2009; Williams et al., 2020; Schoof, 2010), limiting the ability of SMB to project overall mass loss.

Figure 2 caption: centured → centered

Corrected, thank you.

l. 310: "which would be out of the scope of this present study". Still it would be interesting to provide a first order-of-magnitude comparison with RCM produced LIUS SMB.

We removed this section of the manuscript.

l. 332: " XX references"

Thank you for catching that. Fixed!

l.385: "by an error in one of these two estimates". Or in both.

Added, thank you!

---

## Author Comment (AC2)

**Reviewer 2:**

Triggered by the somewhat provokative abstract, I've read the manuscript by Koepnick et al.. To my regret, I adviced the editor that this research does not meet the standards to be publishable in Climate of the Past. Below I'll motivate my advice.

We thank the reviewer for the helpful suggestions. We have made significant revisions to our calculation of the surface mass balance following these suggestions and revised the text accordingly. Specifically, we now account for the ground heat flux, are careful to calculate the SMB using monthly rather annual-average analysis, and have addressed all other comments. These corrections, ground heat flux in particular, had a significant effect on our results, for which we are grateful, and we believe the manuscript has improved considerably.

The final conclusion of the authors is "that General Circulation Models (GCMs) still struggle to reliably calculate the SMB", and this is the correct conclusion for the results presented in the manuscript, but not for ESMs (GCMs is an outdated term) in general. I'm sorry to say that the authors have based their analysis on a model simulation that is not state of the art for modelling the SMB with an ESM. If the authors wanted to know, they could take a look at the recent research by Miren Vizcaíno, for example, to see how well an ESM can model the SMB of an ice sheet, or read the numerous papers by various research groups running coupled atmosphere-ocean-ice sheet models on their extensive efforts to get realistic SMBs within their coupled model environments. In the ITraCE simulation, Brady et al obviously did not. Otherwise they would have calculated the SMB on the fly, but they didn't.

Thank you for your comments. It seems that the issue was not the iTraCE simulation not being state-of-the-art, but rather our neglect of the ground heat flux in our offline calculation. Our conclusions have changed after we corrected our calculation method, and uur calculated SMB is now more consistent with ICE-6G mass rate of change, especially for the early deglaciation period. We followed the work of Miren Vizcaíno, we cite the relevant papers, and exchanged emails with her during the revision to make sure we understood the details of her approach. We note that Vizcaino et al. have used a similar configuration of CESM1 to study the SMB of the GIS. While the reviewer is correct that iCESM1.3 does not include a SMB calculation, we now calculate the SMB offline in a way we feel is satisfactory.

So the conclusion of the manuscript is obvious, namely that if a model doesn't aim to model SMB, it won't get the SMB right. I don't see the need to publish this. It is common sense that if a model is not set up to model a particular property right, there is no chance that by some magic that property will be modelled correctly.

We note that after revising our calculation of the SMB and including the ground heat flux, it is no longer in disagreement with ICE-6G. We also note explicitly in the manuscript the novelty of this study,

> Our goal here is to study the role of SMB during the last deglaciation,

introducing two novel elements to the analysis: First, we calculate the SMB of the LIS continuously in time during the entire deglaciation. Second, we compare the area-integrated SMB to ice mass loss rate deduced from independent geophysical constraints as represented by the ice sheet reconstruction in ICE-6G (Peltier et al., 2015).

The authors give the impression that they have estimated the SMB from the output, and I am willing to believe that the authors think they have estimated the SMB to the best of their ability. The latter, if true, is rather worrying because the manuscript gives the impression that the authors do not know what is essential to model the SMB correctly. This impression arises from the pointless discussion on whether or not to use the geothermal heat flux as an estimate of the ground heat flux of a glaciated surface (the ground heat flux is dominated by other processes),

Thank you for your comments. We now include ground heat flux and calculate the monthly SMB using monthly iTraCE output.

the funny unit error in Figure 1 (if you're talking about kg, you don't need to specify that it's in water equivalents. You do if you're using volume or thickness),

We fixed this typo, thank you for catching that.

the fact that the authors present year-averaged, ice-sheet-averaged fluxes in Figure 2,

we now use annual averages that include only months when melting occurs, thank you for this important comment as well.

and, lastly, the failure to realise that before you start playing around with small changes in albedo, you need to show that albedo makes sense at all. Given that there is negative SMB all the way up to the ice divide, I'd say it doesn't.

We eliminated the albedo sensitivity discussion in the revised manuscript following these suggestions and we feel it is not needed given the revised SMB results.

The authors also seem unaware of the complexity of modelling the SMB of an ice sheet IF you have a proper (online) estimate of meltwater runoff and hence SMB. Ablation zones are generally narrow and steep, especially for land-terminating ice edges

Thank you for mentioning the sensitivity of the narrow ablation zones to model resolution. While we cannot correct the relatively coarse iTraCE resolution, we now acknowledge and discuss this important issue as follows,

> The resolution of the climate model experiment we analyzed is relatively coarse, to enable their long time-integrations. It may not accurately resolve the relatively narrow ablation zones at the ice sheet margins, where steep SMB gradients can occur due to intense localized melting and runoff. This can lead to an underestimation of mass loss, as critical

areas of negative SMB are smoothed out and fail to be fully resolved. Additionally, the coarse atmospheric model resolution may inaccurately represent the interaction between ablation zones and atmospheric processes, such as the advection of warm air and moisture, further biasing SMB estimates. These limitations emphasize the need for higher-resolution models to better resolve the complex dynamics of ablation zones and their contribution to overall ice sheet mass balance.

(ok, the dying glaciers we have around the world contradict this statement, but yes, these glaciers are dying). When working with lower resolution input data, as is the case here, these ablation zones are usually missed, leading to an overestimated integrated SMB. So if you find an SMB that is still too low, then something is really wrong.

With the reviewer's recommended changes, the SMB is now no longer too negative. We hope this means that the resolution of the narrow ablation zones, while still an issue which we acknowledge, still allow us to obtain useful results.

Finally, the authors' assumption that the SMB should at least not be more negative than the long-term mass loss is inadequate.

Given the corrected results, this is no longer an issue. We thank the reviewer again for the helpful guidance.

I assume the authors are aware that the Greenland Ice Sheet, although losing mass, still has a positive SMB of about 30% of its accumulation input.

Yes, thank you.

Yes, the Laurentide Ice Sheet was land-terminating at its southern margin, but marine-terminating everywhere else.

Agreed, we explain now how we carefully deal with grid cells that are part ice, part land (Section 2.5).

Yes, ICE-6G does not provide a mass budget for the Laurentide Ice Sheet, but I don't see what's complicated about investigating the existing modelling literature for the typical mass budget of ice sheet models representing this period, by personal communication if the papers don't reveal it. I'm sorry, but this is poor research practice IMHO.

We have now cited and discussed all existing modeling literature of the simulated SMB of the LIS during the last deglaciation. This includes papers by van den Broeke et al. (2009); Khan et al. (2015); Fyke et al. (2018); Kapsch et al. (2021); Ullman et al. (2015); Bradley et al. (2024); Rutt et al. (2009). Given these, we feel that the novel aspects of our study as identified in our introduction and listed above are valid and would like to think this paper represents a useful contribution. We hope the reviewer agrees given the revision.

So what can be done to improve this manuscript?

In iTraCE, the surface energy balance is not derived for glacier surfaces, or at least not correctly. So simply reconstructing the melt from the surface energy balance (Eq. 1) does not work, as the authors rightly conclude. I don't see the added value of improving the analysis of the results presented here, to conclude again that if a model doesn't try to model SMB, it doesn't model SMB correctly.

As mentioned above, we would put the blame on our previously neglecting the ground heat flux rather than on iTraCE. We now carefully follow papers such as Vizcaíno et al. (2014) and feel the offline SMB calculation provides reliable and useful results, much more consistent with the mass loss rate deduced from ICE-6G.

In my opinion, the authors need to start from scratch. If the authors want to continue to use the iTraCE simulations to estimate the SMB, they need to develop a model or method to estimate the SMB that acknowledges the shortcomings of the GCM data. There are several studies by palaeo-ice sheet modellers using GCM data that have done this before. However, the result of such research is a new paper, not a revised version of this paper.

We agree, have started from scratch, the results have changed significantly. We are embarrassed about having neglected the ground flux previously (having been influenced by some related SMB literature) and we feel the revised manuscript can indeed be considered a new paper...

However, if the editor decides that this manuscript should be given a second chance, I would suggest that

- The authors first discuss the (summer) surface energy balance (including summer albedo and near-surface temperature) and evaluate whether the modelled fluxes are realistic. Modern Greenland and Antarctica can provide some clues as to what might be expected, taking into account orographic and (Milankovisch-driven) isolation differences. In this evaluation, all energy fluxes (including LHF) are expressed in W/m2 and the 'flux convention' is used, i.e. fluxes are positive when directed towards the surface. [And remove the geothermal heat flux, don't embarrass yourself].

Thank you for these suggestions. we

- switched to a monthly surface energy balance and surface mass balance calculation.
- removed geothermal heat flux
- included the ground heat flux
- switched to a convention where all fluxes are positive toward the surface.

- Next, the modelled SMB is analysed.

done.

*- The authors make a realistic estimate of mass loss to the ocean as function of the time, so that a proper "observational" integrated SMB time series is used.*

Hopefully our revised calculated SMB represents a more realistic mass loss estimate. If not, we are admittedly not completely clear on what this item suggests.

*- The authors compare the modeled SMB patterns with SMB patterns used in other studies, generated by other methods.*

Thank you for this comment. We have expanded the discussion of the comparison to existing estimates, including the following text,

> The magnitude and spatial distribution shown in Fig. 4 are similar to Fig. 4 of Kapsch et al. (2021) for the time slices of 15 and 14 ka, based on their simulation of the SMB using a surface albedo evolution model and during the last deglaciation using the Earth system model of the Max Planck Institute (Mauritsen et al., 2019). The negative SMB/net melting (runoff) intrudes into the center of the ice sheet more than in Kapsch et al. (2021). This may be due to the coarser resolution of iTraCE (2° rather than < 1°). Although the spatial distribution of SMB for the LIS was provided by Kapsch et al. (2021) for a few time slices, they only showed the full area-integrated SMB over time for the GrIS, while our focus here is the LIS.

*- When the authors then try to correct the SMB by increasing the albedo, both the new summer albedo fields and the new SMB patterns are shown, at least for some key moments during the transient simulation. The authors should be aware that increasing the albedo decreases the surface temperature, which increases SHF, LHF and LWup (so, e.g. LWup becomes less negative). On instantaneous data, one can make some first order estimates of how large these feedbacks are to arrive at relatively correct updated melt estimates. However, doing this on monthly data is questionable - but ignoring these feedbacks would be even more questionable. So any result of such an analysis should be accompanied by a proper idea of the uncertainties involved.*

We agree and have removed this section. It is no longer needed given that the SMB agrees with ICE-6G now.

*- The authors remove Figure 2 and its subsequent analysis as it stands. You can look at papers such as https://agupubs.onlinelibrary.wiley.com/doi/full/10.1029/2020GL090653 to see how you can work out what processes are driving the melt. I know there is more than one way to partition the energy sources of the melt, but annual, ice-sheet averages are not among the methods that give meaningful results.*

Following this suggestion, Figure 2 has been modified to using monthly data and averaging only over summer months. In any case, this figure (inspired by Figure 6 in https://doi.org/10.5194/cp-20-211-2024). We also added the following to the

caveats section in Ln. 414–416

> We also calculated melt on a monthly time-scale, which may not resolve transient melt events. However, Wang et al. (2021) showed that large melt events contribute little to total surface mass loss for the present-day GrIS.

- The authors revise the wording of their conclusions, acknowledging that models that are not set up to model the SMB cannot be expected to model the SMB correctly. This research doesn't say anything about the performance of GCMs (well, use ESMs, not GCMs) that do try to model the SMB correctly.

The paper has been revised throughout, as suggested. As we acknowledged above, the issue was not so much that the iTraCE model was not set up to model the SMB, but our mistake of not including the ground flux. We take the opportunity to thank the reviewer again for pointing this out.